# AANN: ABSOLUTE ARTIFICIAL NEURAL NETWORK

## ABSTRACT

This research paper describes a simplistic architecture named as AANN: Absolute Artificial Neural Network, which can be used to create highly interpretable representations of the input data. These representations are generated by penalizing the learning of the network in such a way that those learned representations correspond to the respective labels present in the labelled dataset used for supervised training; thereby, simultaneously giving the network the ability to classify the input data. The network can be used in the reverse direction to generate data that closely resembles the input by feeding in representation vectors as required. This research paper also explores the use of mathematical **abs** (absolute valued) functions as activation functions which constitutes the core part of this neural network architecture. Finally the results obtained on the **MNIST dataset** by using this technique are presented and discussed in brief.

## 1 INTRODUCTION

In the field of philosophy, there has been a principle known as 'Ockham's Razor' which, in a simplified relevant language states that "Among the available multiple solutions to the same problem, the simplest one is the best one". For instance, if there are multiple polynomial functions that fit a given data distribution, the lowest degree one would be preferred (Russell & Norvig, 2015). The technique AANN is driven by this principle. In spite of being elementary in its construction, an AANN is able to classify inputs in the forward direction while being able to generate them back in the reverse direction. It can be visualized to be doing classification in the forward direction whereas performing a regression task in the backward direction.

A standalone GAN (Generative Adversarial Network) described in Goodfellow et al. (2014) is able to create representations of the input data by using a novel technique of generating a distribution that contains the original data points as well as data points generated by the Generator part of the network; the distribution is then used by the Discriminator part of the network to classify the data points as genuine or generated. The representations generated by a GAN, although being very effective in creating undistinguishable data points, are however not interpretable and also highly entangled (Chen et al., 2016) (Makhzani et al., 2016). Using an InfoGAN, the problem of entanglement is solved by training in such a way that the network maximises mutual information within small clusters of related latent representations (Chen et al., 2016). Auto-encoder is another technique that uses the concept of encoder-decoder architecture for creating low dimensional representations of the originally very high dimensional input data points. A VAE: Variational Auto-Encoder tries to make the learned representations sparse by using the KL-divergence cost as a regularizer on the final cost of an autoencoder (Kingma & Welling, 2014). Various attempts at combining the two techniques of GAN and VAE have also been made in the unsupervised as well as semi-supervised learning directions (Makhzani et al., 2016) (Larsen et al., 2016). However, these techniques kept getting more and more complicated and somewhere in synthesizing these techniques, it is felt that the 'striving for simplicity' principle has been neglected.

The Absolute Artificial Neural Network exploits all possible information available in the labelled training datasets to structure the learned representations of the input data. Structurally, an AANN is very similar to a feed forward Neural Network with the distinction that AANN uses the **abs** function as the activation function of the neurons. Due to this, all the activations produced, including the hidden layer activations, contain positive real number values. Thus, the network runs on the assumption that the input data as well as the label information comes from a positive data distribution. This doesn't create an issue for the computer vision based tasks. However, for those situations, where

this is not possible, the feature values in the input dataset can be easily moved [1] into the positive region of the multi-dimensional input data space.

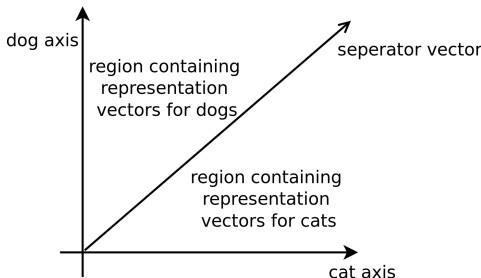

Figure 1: Example of learned representation space created by AANN

The AANN transforms the n-dimensional input data into a space whose number of dimensions are equal to the number of labels used in the training dataset. For instance, presume that, the task is to classify images of cats and dogs and there is a labelled dataset present for achieving this classification. So, the learned representations will contain two dimensions corresponding to each label: cat and dog. The input images are transformed into 2-dimensional vectors by the AANN in such a way that the vectors are as close as possible to their ideal axes. This is achieved by constructing the cost function in a manner that it maximises the *cosine* value of the angle formed by the vector with its ideal axis. As a result, the representation space generated by this AANN can be visualized as shown in the Figure 1. The label axes in the representation space are mutually orthogonal; thus the resulting representation vectors become very interpretable.

## 2 AANN DESCRIPTION

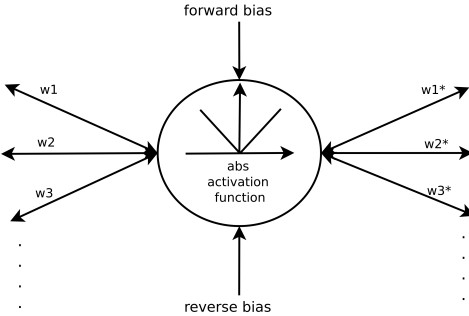

Figure 2: Bidirectional artificial neuron: the building block of an AANN.

The AANN is constructed by using a 'Bidirectional Neuron' (Figure 2) as the building block for the hidden layers of a preliminary feed forward neural network. This bidirectional neuron uses the **abs** (mathematical absolute valued) function as the activation function. The computation performed by the neuron is similar in the forward and the backward directions. In the forward direction, the computation is given by:

$$A_{forward} = \mid (W_{left} * X_{in}) + b_{forward} \mid$$

Whereas, in the backward direction, the neuron computes:

$$A_{reverse} = \mid (W_{right} * X_{in\_rev}) + b_{reverse} \mid$$

---

[1] By mentioning moving the distribution, it is refered to the process of 'change of origin' in cartesian mathematics.

The weights of the hidden layers of the AANN in forward direction learn to compute a function for transforming the input data into the representation vectors. While in the reverse direction, the weights constitute a function for constructing data points that closely resemble the data points belonging to the input dataset from the representation vectors. It is highly intriguing, and at the same time enigmatic, that the same set of weights constitute two entirely distinct functions.

## 2.1 FORWARD PASS

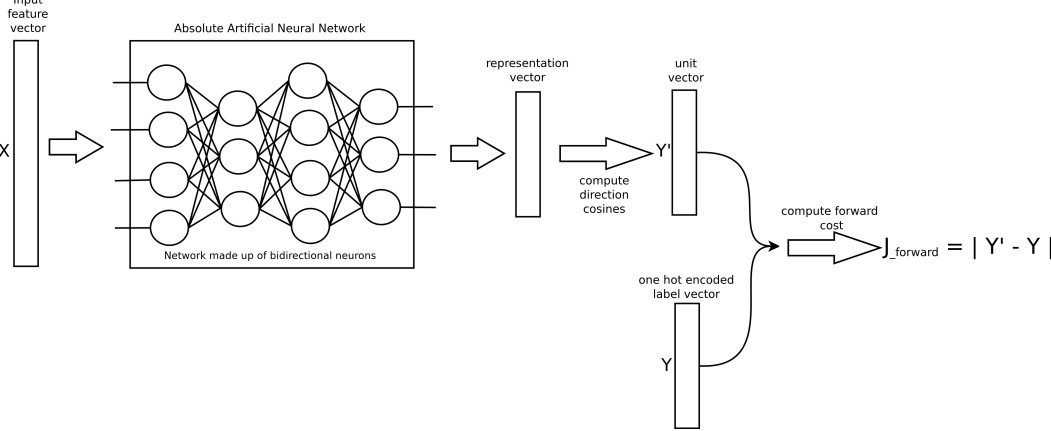

Figure 3: Forward pass of the AANN.

The input n-dimensional feature vector is passed through the neural network consisting of hidden layers, constructed from the bidirectional neurons, to obtain an m-dimensional representation vector; where m corresponds to the number of labels. The obtained representation vector is then converted into a unit vector, which primarily corresponds to the cosines of the angles made by the representation vector with the coordinate axes. Finally, the forward cost $J_{forward}$ can be computed as either the Euclidean distance or just the mean absolute difference, which is an estimate of the euclidean distance, between the unit representation vector $Y'$ and the one-hot-encoded-label vector $Y$.

The *direction cosines* of the vector can be obtained by using the formula:

$$\text{if } A = [x_1, x_2, x_3, ..., x_m]; \text{ then } \mid \overline{A} \mid = \sqrt{x_1^2 + x_2^2 + x_3^2 + ... + x_m^2}$$
$$A_{cosine} = \frac{\overline{A}}{\mid \overline{A} \mid}$$

i.e. by scaling every activation value present in the representation vector by the inverse of the magnitude of the vector. This results in a unit vector that only corresponds to the direction of the original vector. As per the forward cost, it is intended to bring this direction vector as close as possible to the ideal label coordinate axis. Due to which, the label axis encodes the input information as representation vectors of different magnitudes converge on it. [*link*] [2] This visualization demonstrates how information gets encoded along the label axis in various real valued magnitude ranges. The visualization was generated by interpolating a small of range of values, precisely $[0 - 100)$, along all 10 different axes corresponding to the 10 digits, present in an **MNIST** dataset, in a sequence by using a trained AANN. It is clearly evident from the visualization that the network creates more than just input output mappings; it creates a function of the learned representations as apparent from the smooth transitions between the different forms of a digit along it's dedicated axis.

---

[2]Since this is a submission for the double-blind review, the link has been redacted not to reveal any identities.

## 2.2 REVERSE PASS

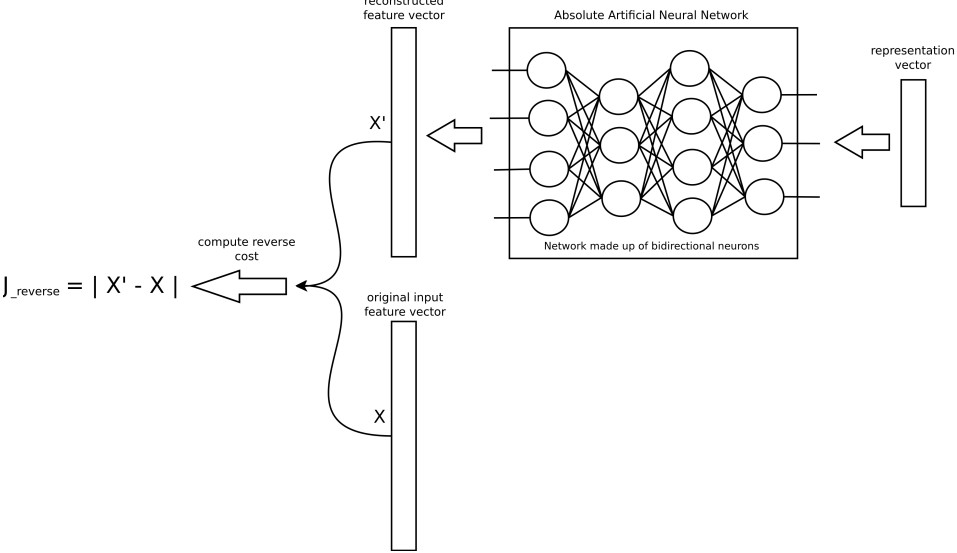

Figure 4: Reverse pass of the AANN.

During the reverse pass of the AANN, the representation vector emitted by the network of hidden layers in the forward pass is fed back into the network in the reverse direction [3]. The network then performs transpose operations to give off a new vector $X'$ in the input n-dimensional space. The reverse cost $J_{reverse}$ is computed as either the euclidean distance or the mean absolute difference between the vectors $X'$ and $X$. By defining the reverse cost in such a way, it is intended to obtain the vector $X'$ as close as possible to the original input vector $X$. This accords the network the ability to generate data points in the input space in the reverse direction.

## 2.3 TRAINING:

The network is trained by using the Backpropagation (Rumelhart et al., 1986) algorithm to minimise the final cost $J_{final}$. The final cost is defined as the sum of the forward and the reverse costs.

$$J_{final} = J_{forward} + J_{reverse}$$

It is ultimately this cost with respect to whom the partial derivatives of the parameters are computed. The parameters are then adjusted by using the computed derivatives according to the Adam optimization as described in Kingma & Ba (2017).

This action of performing the forward pass to calculate the forward cost followed by the reverse pass to obtain the reverse cost and then performing backpropagation on the final cost constitutes a single pass of the AANN. The term AANN: Absolute Artificial Neural Network, which is also the title of the paper, thus refers to this unified process of training a neural network in such a way.

## 3 EXPERIMENTATION WITH OTHER ACTIVATION FUNCTIONS

This section attempts to succinctly describe the process of, and findings attained by, using other activation functions for the neural network architecture described in the previous section. Since the actual reasons why these activation functions behave in the manner that they do are not fully known, it has been tried to remain fatihful while describing the experiments and not to make any unproven, or otherwise philosophical, remarks in this section. The programming implementations of these experiments have been made available at [*link*].

---

[3]Note that the unscaled representation vector and not the directional cosine vector (Y' in Figure 3) is fed back.

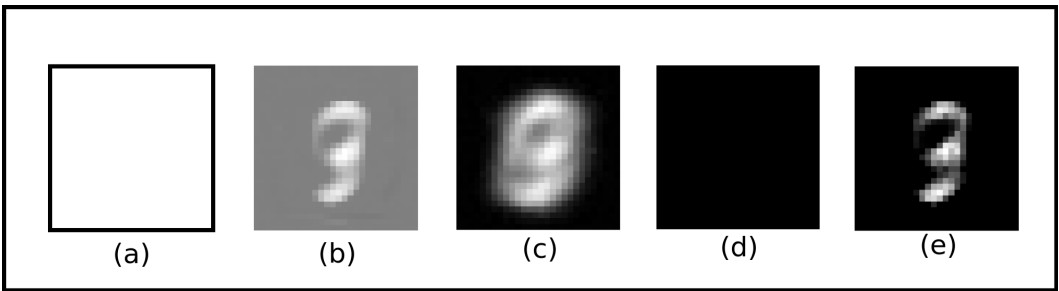

Figure 5: Images generated in the reverse direction by different activation function settings of an AANN. (a) Use of ReLU activation function. (b) Linear activation function. (c) ReLU in the forward direction and Abs in the backward direction. (d) Abs forward and ReLU backward. (e) Use of Sigmoid activation function.

Upon using the ReLU, i.e. Rectified Linear-Unit, function (Nair & Hinton, 2010) as the activation function for this architecture, all the activations shoot to *nan* [4] in the forward direction leading to proliferation of *nan* in the reverse direction as well. If the Linear activation function is used, the network performs poorly in the forward direction, leading to very high classification error rates, while, the network converges to the point that it outputs the same structure as shown in *(b)* of Figure 5 for every possible representation vector. On activating the hidden neurons with a ReLU in the forward direction and with an Abs in the reverse direction, the network kills all the activations, i.e. outputs the zero vector for every input, in the forward direction. In the backward direction, the network converges to the *(c)* structure. Upon using the Abs function in the forward direction and the ReLU in the backward direction, the network this time kills all the activations in the backward direction as visualized in *(d)*. The *(e)* in Figure 5 is the output achieved by using the Sigmoid activation function in the network. The result obtained is very similar to the result of using Linear activation function, as in *(b)*.

## 4    RESULTS ON MNIST DATASET

The AANN architecture was trained on the MNIST digit recognition dataset[5]. The dataset contains [(28 x 28) pixels] sized images of handwritten digits from 0 - 9. The programming implementation using the Tensorflow framework (Abadi et al., 2015) has been made available at [*link*].

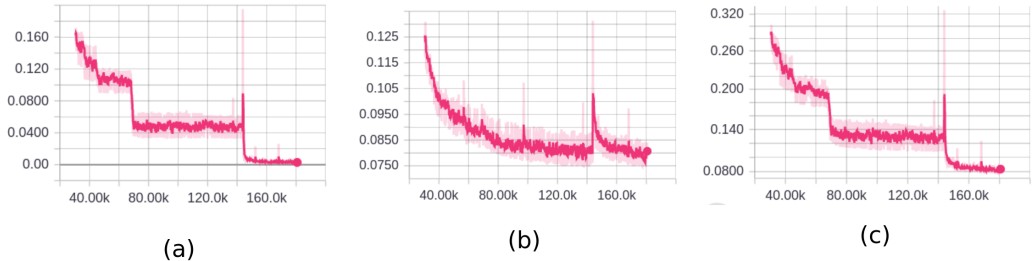

Figure 6: Cost plots obtained upon training the AANN on the MNIST digit dataset. (a) Forward cost. (b) Reverse cost. (c) Final cost.

There are 42000 images in the training set, of which, 95% were used for train set and remaining 5% images were used for the dev set. i.e. 39900 in the train set and 2100 in the dev set. The network was trained using the Adam (Kingma & Ba, 2017) optimizer with $\alpha = 0.001$, $\beta_1 = 0.9$, $\beta_2 = 0.999$ and $\epsilon = 10^{-8}$.

---

[4]nan: 'Not A Number'; used in the programming terminologies.
[5]https://www.kaggle.com/c/digit-recognizer

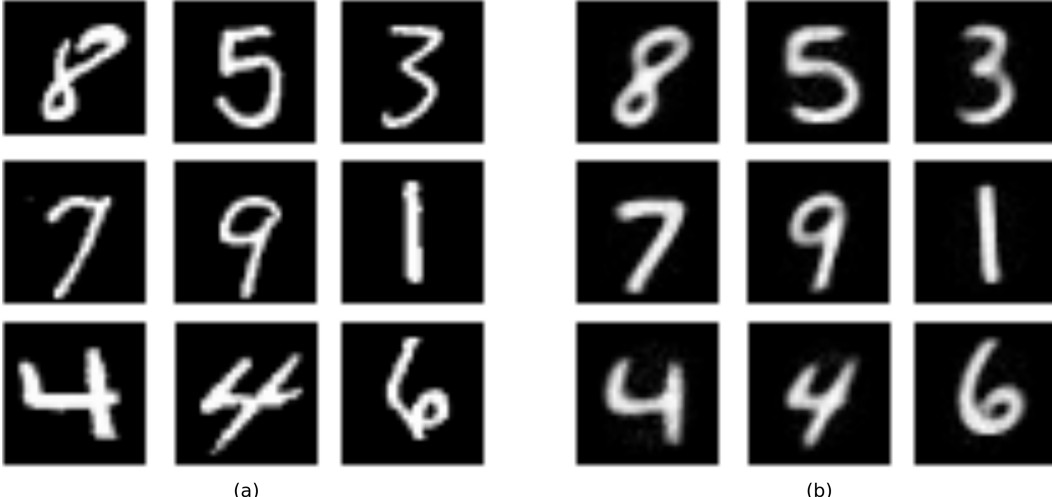

(a)                 (b)

Figure 7: Outputs generated by the AANN in the reverse direction. (a) Original images fed into the network. (b) Images reconstructed by the network in the reverse direction.

The network achieved a classification accuracy score of **99.86%** on the train set and **97.43%** on the dev set in the forward direction. The unseen test set of this version of the dataset used contains another **28000** images for which the network achieved an accuracy of **97.671%**. Figure 7 shows the images generated by the network in the reverse direction against the original images fed to the network. It is perceived that the capability of the network should not be evaluated only on the basis of it's forward accuracy scores but should be evaluated on the basis of a unified metric that not only measures the network's forward performance but also the faithfulness with which the network is able to generate input data points in the reverse direction.

## 5 CONCLUSIONS AND FUTURE SCOPE

This research paper put forth an elementary but potent neural network architecture, named as AANN, that has the ability to learn in the forward as well as the backward direction. It also proposed the Abs function as a viable activation function for a neural network architecture. Due to lack of hardware resources, the experimentation had to be limited to the preliminary MNIST dataset, but it is firmly believed that the technique will perform equally well upon tackling other robust datasets, because of the theoretical evidence shown in the performed experiments.

The AANN presently encodes the information in real number valued ranges across the the dedicated label axes in the the representation space. Certain regularization functions can be synthesized in order to stretch these ranges so that more information can be incorporated in them. The number of dimensions of the learned representations can be manually controlled by setting certain number of dedicated axes to a single label and by modifiying the forward cost function in such a way that the representation vectors lie inside the space generated by the coordinate axes dedicated to the ideal label. An in depth mathematical study of the Abs activation function could reveal the underlying behaviour of AANN. This forms the future scope for research.

This technique also opens up new research opportunities for considering the AANN architectural modifications to certain network architectures like Rasmus et al. (2015) for semi-supervised learning. Moreover, it would be interesting to note the implications of applying the corresponding modifications to more advanced architectures such as Conv-nets (Krizhevsky et al., 2012) and Recurrent Nets with LSTM cells (Hochreiter & Schmidhuber, 1997).

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
