# OpenReview forum: "AANN: Absolute Artificial Neural Network"
_ICLR.cc/2018/Conference — Reject_

### Official Review · AnonReviewer2 · 2017-11-25
**Preliminary report on a classification/reconstruction network with absolute value activation function**

**Rating:** 2
**Confidence:** 3

**Review:**

SUMMARY

The model is an ANN whose units have the absolute value function abs as their activation function (in place of ReLU, sigmoid, etc.). The network has bi-directional connections (with equal weights) between consecutive layers, but it operates only in one direction at a time. In the forward direction, it is a feed-forward net from image to classification (say); in the reverse direction, it is a feed-forward net from classification to image. In both directions it operates in supervised fashion, trained with backpropagation (subject to the constraint that the weight matrix is symmetric). In the forward direction, the activation vector y over the classification layer is L2-normalized so the activation of a class c is the cosine of the angle between y and the 1-hot vector for c.
Although there is a reverse pass through the net, the training loss function is not adversarial; the loss is just the classification error in the forward pass plus the reconstruction error in the backward pass.
The generalization accuracy in classification on 42k-image MNIST is 97.4%.

STRENGTHS

* Comparisons are made of the reconstruction performance with the proposed abs activation function and with ReLU on one or both passes, and a linear activation function.
* The model has the virtue of simplicity, as the authors point out.

WEAKNESSES

* The discussion of evaluation of the model is weak.
  - No baselines are given. (A kaggle leaderboard shows the 50th-ranked model at 97.8% and the top 8 models at 100%.)
  - The paper talks of a training set and a "dev" set, but no test set, and generalization performance is given for the dev set rather than a test set.
  - No quantitative evaluation of the reconstruction (backward pass) performance is given, just by-eye comparison of the reconstruction error through figures.
  - Some explanation is needed of why the ReLU cases were fatally plagued with NaN errors.
* Claims of interpretability advantage seem unwarranted since the claimed interpretability applies to any classification ANN, as far as I can see.
* The work seems to be at too preliminary a stage to warrant acceptance at ICLR.

---

> ### Author Response · Authors · 2018-01-06
> **Clarifications**
>
> Thank you for the review.
>
> I would like to address the weaknesses that have been pointed out and try to give an explanation for them.
>
> 1.) The choice of a baseline was a bit unclear, since all the recorded models present on the MNIST leaderboard only perform classification and do not have the reconstruction module through the same network. Besides, I perceive that comparing just the forward performance, as I have mentioned in the paper, is a bit unfair in this case.
>
> 2.) The new revision of the paper has now included the details about the test set results.
>
> 3.) I have presented a by eye comparison since quantitatively measuring the likeness among the reconstructed images and the original images is mathematically challenging, whilst also being susceptible to the pixel level difference in the noise smoothing caused by the reconstruction network. It has been mentioned in the paper, that a metric that takes into consideration all these factors while evaluating the backward performance faithfully is needed.
>
> 4.) The mention of interpretable encodings (representations) is made since through this free normalization loss function, all the information about the digit's positions, orientations, thickness and curvedness is summerized along a positive real number range.
>
> Thank you.

---

### Official Review · AnonReviewer3 · 2017-11-27
**incremental idea, insufficient experimental evaluation**

**Rating:** 3
**Confidence:** 5

**Review:**

The paper proposes using the absolute value activation function in (what seems to be) an autoencoder architecture with an additional supervised learning term in the objective function that encourages the bottleneck layer representation to be discriminative. A few examples of reconstructed images and classification performance are reported for the MNIST dataset.

The contribution of the paper is not clear. The idea of combining autoencoders with supervised learning has been explored before, see e.g., "Learning Deep Architectures for AI" by Bengio, 2009, and many other papers. Alternative activation functions have also been studied in many papers, see https://arxiv.org/pdf/1710.05941.pdf for a recent example. Even without novel algorithmic contributions, the paper would have been interesting if there was an extensive evaluation across several challenging datasets of different ways of combining autoencoders with supervised learning and different activation functions that gives better insight into what works and why.

It would be helpful not to introduce new terminology like "bidirectional artificial neuron" unless there is a significant difference from existing concepts. It is not clear from the paper how a network of bidirectional neurons is different from an autoencoder.

---

> ### Author Response · Authors · 2018-01-06
> **Clarifications**
>
> To state explicitly what the intended features are:
>
> 1.) Use of abs function (which has been duly noted)
>
> 2.) Use of free normalization in objective function definition. It has been seen that usually the loss (objective) function is defined not only such that the prediction is close to the actual label but also so that the probabilities of other labels are minimised. My proposal is to let the network only focus on getting the correct label right while the latter is taken care of automatically if reconstruction is to be done through the same network.
>
> 3.) The hypothesis that the process of reconstruction should be symbiotic with the process of classification / prediction and not adversarial. In the paper (Sabour et. al. 2017) the CapsNet uses a separate fully connected reconstruction module and uses the reconstruction loss as a regularizer similar to the technique described in this paper. By simply summing the reconstruction loss with the objective function, the process of learning becomes more symbiotic.
>
> 4.) From the visualization of the reconstructed digits from the encodings, it can be seen that the forward classification function just doesn't learn discrete mappings of input - output pairs, but learns a smooth function that encodes different positions, orientations, thickness and curvedness of the input digits along a simple positive real number range.
>
> 5.) All these code implementations and visualization videos are there, but I couldn't mention them in the paper due to the anonymity clause.
>
> 6.) The intention of using the term 'bidirectional artificial neuron' was to give a simpler perspective at an Autoencoder. It is not different from an Autoencoder, it is merely a simpler explanatory view of it which I put forth through the article.
>
> Thank you for the reviews. All the comments are very helpful and strengthen my further work.

---

### Official Review · AnonReviewer1 · 2017-11-28
**Interesting ideas but not fully explored**

**Rating:** 6
**Confidence:** 4

**Review:**

This paper introduces a reversible network with absolute value used as
the activation function.  The network is run in the forward direction
to classify and in the reverse direction to generate.

The key points of the network are the use of the absolute value
activation function and the use of (free) normalization to match
target output. This allows the network to perfectly map inputs to any
point on a vector that goes through the one-hot encoding, allowing for
deterministic generation from different vectors (of different lengths)
with the same normalized output.

I think there are a lot of novel and interesting ideas in this paper
though they have not been fully explored.  The use of the absolute
value transfer function is new to me, though I was able to find a couple of old
references to its use.   In a paper by Gad et al. (2000), it is stated
" For example, the algorithm presented in Lin and
Unbehauen (1995) < I think they mean Lin and Unbehauen 1990)>
 is used to train networks with a single hidden layer
employing the absolute value as the activation function of the hidden
neuron. This algorithm was further generalized to multilayer networks
with cascaded structures in Batruni (1991)."   Exploring the properties
of the abs activation function seems worth exploring.

More details on the training are needed for full clarity in the paper.
(Though it is recognized that some of these could be determined from
links when made active, they should be included in the paper).  How
did you select the training parameters given at the bottom of page 5?
How many layers and units/layer did you use? And how were these
selected?  (The use of the links for providing code and visualizations (when active)
 is a nice feature of this paper).

Also, did you compare to using the leaky ReLU activation function --
That would be interesting as it also doesn't have any areas of zero
slope?  Did you compare the generated digits to those obtained using GANs?

I am also curious, how does accuracy on digit classification differ
when trained only to optimize the forward error?

The MNIST site referenced lists 60,000 training data and test data of
10,000.  How/why did you select 42,000 and then split it to 39900 in
the train set and 2100 in the dev set?

Also, the goal for the paper is presented as creating highly
interpretable representations of the input data.  My interpretation of
interpretable is that the hidden units are "interpretable" and that it
is clear how the combined hidden unit representations allow for
accurate classification.  Towards that end, it would be nice to see
some of the interpretations of the hidden unit representations.  In
the abstract it states " ...These representations are generated by
penalizing the learning of the network in such a way that those
learned representations correspond to the respective labels present in
the labelled dataset used for supervised training".  Does this
statement refer only to the encoding of the representation vector or
also the hidden layers?  If the former, isn't that true for all
supervised algorithms.  If the latter, you should show this.

Batruni, R. (1991). A multilayer neural network with piecewise-linear
structure and backpropagation learning. IEEE Transactions on Neural
Networks, 2, 395–403.

Lin, J.-N., & Unbehauen, R. (1995). Canonical piecewise-linear neural
networks. IEEE Transactions on Neural Networks, 6, 43–50.

Lin, J.-N, & Unbehauen, R. (1990). Adaptive Nonlinear Digital Filter with Canonical Piecewise-Linear Structure,
IEEE Transactions on Circuits and Systems, 37(3) 347-353.

Gad, E.F et al (2000). A new algorithm for learning in piecewise-linear neural networks.
Neural Networks 13,  485-505.

---

### Decision · Program_Chairs · 2018-01-29
**ICLR 2018 Conference Acceptance Decision**

**Decision:**

Reject

**Comment:**

The paper proposes to use absolute value activations, in a joint supervised + unsupervised training (classification + deep autoencoder with tied encoder/decoder weights).
Pros:
 + simple model and approach on ideas worth revisiting
Cons:
- The paper initially approached these old ideas as novel, missing much related prior work
- It doesn't convincingly breathe novel insight into them.
- Empirical methodology is not up to standards (non-standard data split, lack of strong baselines for comparison)
- Empirical validation is too limited in scope (MNIST only).